# Salt-Containing Recipes in Popular Magazines with the Highest Circulation in the United States Do Not Specify Iodized Salt in the Ingredient List

**DOI:** 10.3390/ijerph20054595

**Published:** 2023-03-05

**Authors:** Josephine Uerling, Emily Nieckula, Katarina Mico, Arantxa Bonifaz Rosas, Emmie Cohen, Helena Pachón

**Affiliations:** 1Emory College of Arts and Sciences, Emory University, Atlanta, GA 30322, USA; 2Rollins School of Public Health, Emory University, Atlanta, GA 30322, USA

**Keywords:** iodine, fortification, iodization, consumption

## Abstract

Iodine deficiency is a public health problem in the US, with the iodine status of women of reproductive age decreasing in recent years. This may be attributable to voluntary salt iodization in the US. Magazines, a common source of recipes and nutritional information, may influence salt use and iodine intake. The aim of this study is to assess whether the magazines with the highest circulation in the US include recipes that contain salt and, if so, whether they specify “iodized salt” in the recipes. Recipes in eight of the top ten magazines by circulation in the US were examined. Standardized information was collected on the presence and type of salt in recipes in the last 12 issues reviewed per magazine. About 73% of the 102 issues reviewed contained recipes. A total of 1026 recipes were surveyed for salt; 48% of the recipes listed salt as an ingredient. None of the 493 recipes containing salt specified iodized salt as the type of salt to be used. About half of the recipes in the last 12 issues of popular magazines published in the US included salt in the ingredient list; however, none recommend the use of iodized salt. There is potential for editorial changes among magazines to call for iodized salt in recipes, which may further prevent iodine deficiency in the US.

## 1. Introduction

Iodine deficiency disorders, while not as prevalent in the US as in other regions worldwide [1], are a significant and preventable public health problem in the US. Inadequate iodine intake, which is represented by a median urinary iodine concentration (UIC) of less than 100 μg/L in children over six years and most adults and less than 150 μg/L in pregnant women [2], can lead to health consequences such as goiter, hypothyroidism, and mental impairments [3]. Pregnant women are particularly vulnerable due to the role of iodine in neonatal neurological development [4], making iodine deficiency a significant health concern for women of reproductive age (WRA). The iodine recommended dietary allowance is 220 μg per day for pregnant women and 150 μg for non-pregnant women [5], with common sources of iodine including seafood, dairy products, and iodized salt [6]. According to a 2018 review that compiled National Health and Nutrition Examination Survey data with data from additional studies on iodine status in the US, iodine levels in non-pregnant WRA in the US have decreased from 1970 to 2012 [7]. In 2017, the Iodine Global Network global scorecard reported a median UIC of 129 μg/L in pregnant women in the US [8], which is below the cutoff for an insufficient median UIC.

Salt iodization is a common and effective intervention to reduce iodine deficiency, with 126 countries currently mandating the fortification of salt with iodine [9]. Although the WHO recommends universal salt iodization [10], there are 21 countries, including the US, in which salt iodization is voluntary [9]. In 2015, approximately 53% of salt sold in the US retail market was iodized [11], indicating that a significant portion of the population in the US may not be consuming enough iodine through salt. Despite this, salt iodization remains voluntary in the US, and there are still concerns about iodine deficiency in some groups. Research on the effectiveness of salt iodization, in both mandatory and voluntary settings, has consistently found that iodized salt significantly reduces the risk of various iodine deficiency disorders and is associated with an increase in median UIC [12,13].

To our knowledge, there have been no studies in the US on the knowledge, attitudes, and behaviors related to iodine or iodized salt. However, other studies in high-income countries have reported low levels of iodine knowledge among participants. A study in the UK found that only 12% of participants were aware of iodine intake recommendations and 16% could identify the impacts of iodine deficiency [14], and a study in Norway found that 53% of pregnant women did not know any health outcomes related to iodine [15]. Only 26.4% of university students in six countries in Europe and Asia were aware of the relationship between iodine deficiency and poor cognitive development [16]. Similarly, only 43% of WRA in Australia could identify cognitive outcomes related to low iodine status [17]. Awareness of iodized salt as a source of iodine is typically low as well, with 41% of pregnant women surveyed in Norway and less than half of participants from Germany, Greece, Poland, and Slovenia identifying iodized salt as a source of iodine [15,16].

With over 200 million readers per year in the US from 2012 to 2019 [18], magazines are an accessible and common source of information for North Americans. Previous research has found that magazines are a popular and influential source of recipes and nutrition information [19,20,21,22]. The specific ingredients used in magazine recipes may influence what ingredients magazine readers use, and, therefore, the inclusion of ingredients such as iodized salt in magazine recipes may influence iodine consumption. Despite limited data regarding what percentage of magazine readers actually use the ingredients specified in magazine recipes, other studies have found that safety information included in recipe instructions promotes food safety behaviors [23,24].

WRA are frequent readers of magazines in the US, with a 2013 survey finding that 84% of affluent women reported reading at least 1 of 135 print magazines and 52% of affluent women, compared to 23% of men, reported reading cooking magazines [25]. Additionally, in a 2020 survey, 36% of women ages 18 to 64 reported reading more magazines during the COVID-19 pandemic [26]. Therefore, magazine recipes, and by extension the ingredients included in them, are frequently viewed by women, the population most at risk for iodine deficiency disorders.

Previous studies have investigated salt use, micronutrient information, and other nutritional content in magazine recipes, advertisements, and articles [27,28,29,30,31,32,33,34,35,36]; however, none have focused on iodized salt in magazine recipes. Some of this earlier research has found that nutritional content in magazine recipes does not align with nutritional recommendations [27,30,31,36]. Our objective is to determine whether the magazines with the highest circulation in the US include recipes that contain salt and, if so, whether they specify “iodized salt” in the recipes.

## 2. Materials and Methods

The research consisted of a systematic survey of the highest-circulation magazines in the US from December 2019/January 2020 to November 2021.

### 2.1. Selection of Magazines

The top ten magazines with the highest circulation in the US as of June 2021 were identified by the Alliance for Audited Media [37]. We aimed to analyze the inclusion and description of salt in recipes in the top ten magazines. The magazines were accessed online through the Emory University Library and local libraries in Atlanta. However, two of the top ten magazines were not available (*Allrecipes* and *WebMD Magazine*); thus, they were excluded from our analysis. The magazines included in this study were *AARP the Magazine*, *Better Homes and Gardens*, *Cosmopolitan*, *Good Housekeeping*, *People*, *Taste of Home*, *US Weekly*, and *Vanity Fair*.

### 2.2. Data Collection

For each magazine, we intended to review and analyze the recipes included in the 12 most recent issues, excluding recipes that were part of advertisements. If an issue did not contain any recipes, we reviewed an additional issue of that magazine to find 12 issues that did include recipes. We excluded a total of nine issues due to unavailability at libraries or electronically. Upon completion, 102 issues from eight magazines were analyzed, with issue dates ranging from December 2019/January 2020 to November 2021 (Table 1).

For each recipe, the magazine, issue date, recipe name, whether it contains salt, and the type of salt were recorded. Only the ingredients listed for each recipe were reviewed and recorded. If a recipe mentioned salt in the directions but not in the ingredient list, we determined that the recipe ingredient list did not contain salt. Two researchers independently scored each recipe for reliability.

### 2.3. Data Analysis

For each magazine, the number of issues with recipes, number of recipes, number of recipes containing salt, and number of salt-containing recipes stipulating “iodized salt” were tabulated and summarized using Google Sheets (Appendix A). Additionally, we compared the types of salt and calculated the number of salt varieties listed in the ingredient list of eight magazines.

## 3. Results

Among the eight magazines reviewed, there were a total of 102 issues reviewed (Table 1). *US Weekly* had the most issues assessed (23.53%). Seventy-four of the issues examined contained recipes (Table 2). Of the 13 *Vanity Fair* issues reviewed, none contained a recipe.

There were a total of 1026 recipes identified with issue dates ranging from December 2019/January 2020 to November 2021 (Table 2). Salt was listed as an ingredient in 493 of the recipes. Of these salt-containing recipes, none listed iodized salt as the type of salt to be used.

There were 14 different types of salt included in the ingredient list in the magazines reviewed (Table 3). About 52% of the recipes containing salt did not specify what type of salt to use. Kosher salt was the most recommended salt type among recipe ingredient lists (41.09%). One recipe stated that the reader could choose to cook with either celery salt or garlic salt. This resulted in one additional count in the number of recipes by types of salt listed compared to the number of recipes containing salt (Table 2 and Table 3).

When comparing the salt types by magazine, *Taste of Home* listed the most types of salt (*n* = 10) in the recipes reviewed, followed by *Better Homes and Gardens* (*n* = 7) (Table 4). The 218 recipes from *Cosmopolitan* and *Good Housekeeping* did not specify salt type and only listed salt in the ingredient list.

## 4. Discussion

In the 12 latest issues among eight of the ten leading magazines in the US, none stipulated “iodized salt” in the ingredient list for 493 recipes. The lack of emphasis on the use of iodized salt in magazine recipes presents a challenge to the health of WRA in the US, whose iodine levels have decreased over the last few decades [7]. Iodized salt is one of the main sources of iodine for the US population [38]. The decrease in iodine levels can lead to iodine deficiency, which is a public health concern due to its adverse health outcomes. In the US, iodine deficiency has been minimized in the past through the introduction of voluntarily iodized salt to consumers [38]. The potential re-emergence of iodine deficiency among WRA in the US can cause detrimental health outcomes in women and their children, consequently affecting their economic productivity and quality of life [3,4].

To our knowledge, there are no other studies analyzing magazine recipes for the inclusion of iodized salt or other fortified foods. One study of the ten most circulated magazines in Australia assessed recipes for salt content and compliance with national guidelines for salt consumption [27]. While our study focused on the description of salt and the specification of iodized salt, the study in Australia analyzed salt levels and directions for salt use in different recipes. Webster et al. found that a majority of recipes included high-salt ingredients, with salt use that did not align with the Australian national dietary guidelines. In relation to these previous findings, our results further suggest that directions for salt use in magazine recipes do not reflect nutritional recommendations, in particular, the consumption of iodine.

Several other studies have analyzed nutritional content and claims in recipes, articles, and advertisements in popular magazines, with two looking at micronutrients other than iodine [28,29] and one, in addition to the work of Webster et al. [27], including sodium content in its analyses [30]. Two studies analyzed other nutritional content [31,32], and five focused on nutritional claims and messages rather than the specific nutritional content of recipes. Like our study, these investigations searched popular magazines for specific nutritional content and analyzed them in relation to nutritional recommendations.

The number of magazines analyzed ranged from one [31,33,34] to sixteen [32] with an average of 4.7 magazines among the ten other studies reviewed. Similar to our study, five others chose magazines based on widespread readership [27,29,30,32,35]. There was also variation in the type of magazine content analyzed for nutritional information, with some studies looking at recipes [27,30,31] and others assessing advertisements [28,33,34,35,36] or articles [28,29,32]. Overall, few studies have analyzed magazine recipes specifically, and none have looked at references to iodized salt or other fortified foods in magazine recipes.

Several of these previous studies found that nutritional content in magazines often did not align with nutritional recommendations [27,30,31,36]. In contrast, one study found that nutritional claims in US magazines reflected dietary recommendations [35], and two found that Canadian magazines included positive health effects of micronutrients in advertisements and articles [28,29]. Our results alongside the findings of similar studies indicate variation in how nutritional information is included in magazine content; however, our study suggests that there is a lack of information about iodized salt in popular magazines.

Kosher salt emerged as the most cited salt in the recipes we reviewed. In this research, we did not survey consumers about the salt(s) they choose. However, in a letter to the editor, Shader stated that “[kosher salt] is preferred because its larger and rougher granules draw out the juices from meats and poultry and they are easier to spread” [39].

The simultaneous growing consumption of plant-based products and decrease in dairy consumption in the US pose a risk of increasing iodine deficiency in the population. The US Department of Agriculture reported that from 2013 to 2017, household weekly average purchases of cow’s milk declined by 12% [40]. On the other hand, purchases of almond, soy, and other plant-based “milk” products increased by 36%. One potentially impactful strategy to combat insufficient iodine intake in the US population would be to mandate salt fortification with iodine as is currently legislated by 126 countries [9].

Efforts to increase iodine consumption through iodized salt are compatible with salt intake reduction strategies to reduce the intake of sodium and concomitant risk of blood pressure and cardiovascular diseases [41]. Essentially, as salt intake declines, iodine levels in salt can be increased (and vice versa). Communication with the general public, and particularly WRA, is needed to promote both the importance of iodine intake and reduced salt consumption.

### Strengths and Limitations

Although other studies have looked at salt use [27], micronutrient information [28,29], and sodium content [30] in magazines, our study is the first to investigate the presence of iodized salt, or any fortified food, in magazine recipes. This is significant due to the growing problem of iodine deficiency in women in the US and the potential health risks of insufficient iodine intake. Additionally, our study is strengthened through the selection of magazines based on high circulation. Our use of eight different magazines—relative to the average of 4.7 magazines in previous studies reviewed—also benefits our analysis, accounting for differences between magazines as well as variation in readership.

Multiple researchers collected data for this analysis, thus introducing potential biases and imprecision in data collection. We aimed to minimize inconsistency in data collection at the beginning of the study by completing a standardization exercise, and two individuals independently scored the same recipes to ensure reliability. Limited generalizability was also a limitation of our study, as we included the latest twelve issues prior to October 2021 of popular magazines selected based on high circulation in the US. The number of recipes per magazine was highly variable as some magazines had a greater number of recipes per issue than others. Moreover, most of the magazines analyzed were not cooking magazines, which would limit the generalization of our results to all magazine recipes. As observed, the cooking magazine *Taste of Home* included the highest number of salt types in their recipes. We were unable to access issues from two of the most popular magazines (*Allrecipes* and *WebMD Magazine*), which limited our sample size and may have influenced our findings.

## 5. Conclusions

We reviewed nearly 500 salt-containing recipes in 74 issues of eight magazines with high circulation in the US. None of the recipes specified iodized salt in the ingredient list. Based on our results, we recommend magazine editors specifically call for iodized salt in the food recipes of US magazines. A greater emphasis by the media on the use of iodized salt in food preparation could increase the iodine intake in WRA and prevent the re-emergence of iodine deficiency in the US. Since adherence to the ingredients that recipes call for is not always guaranteed, it would be valuable for recipes to include a note that emphasizes the benefits of using iodized salt. Both editorial changes could encourage the magazines’ female subscribers to incorporate iodized salt into their diet, which could prevent them from developing iodine deficiency.

## Figures and Tables

**Table 1 ijerph-20-04595-t001:** Issues and dates of each magazine reviewed for inclusion of salt and iodized salt in the list of ingredients in magazines published between 2019 and 2021.

Magazine	Number of Issues Reviewed	Number of Excluded Issues *	Earliest Date Reviewed	Latest Date Reviewed
*AARP the Magazine*	7	5	December 2019/January 2020	August/September 2021
*Better Homes and Gardens*	12	0	October 2020	September 2021
*Cosmopolitan*	14	0	June 2020	November 2021
*Good Housekeeping*	12	0	July/August 2020	September 2021
*People*	12	0	26 July 2021	11 October 2021
*Taste of Home*	8	4	August/September 2020	October/November 2021
*US Weekly*	24	0	31 May 2021	2 November 2021
*Vanity Fair*	13	0	October 2020	November 2021
Total	102	9	December 2019/January 2020	November 2021

* Unavailable electronically or at libraries.

**Table 2 ijerph-20-04595-t002:** Numbers of issues and recipes reviewed for inclusion of salt and iodized salt in the list of ingredients in magazines published between 2019 and 2021.

Magazine	Number (%) of Issues	Number (%) of Issues with Recipes	Number (%) of Recipes	Number (%) of Recipes Containing Salt	Number of Salt-Containing Recipes Stipulating “Iodized Salt”
*AARP the Magazine*	7 (6.68%)	7 (9.46%)	36 (3.51%)	9 (1.83%)	0
*Better Homes and Gardens*	12 (11.76%)	12 (16.22%)	223 (21.73%)	32 (6.5%)	0
*Cosmopolitan*	14 (13.72%)	13 (17.58%)	41 (4%)	21 (4.3%)	0
*Good Housekeeping*	12 (11.76%)	12 (16.22%)	264 (25.73%)	197 (40%)	0
*People*	12 (11.76%)	12 (16.22%)	25 (2.43%)	22 (4.5%)	0
*Taste of Home*	8 (7.84%)	8 (10.81%)	410 (40%)	206 (41.8%)	0
*US Weekly*	24 (23.53%)	10 (13.51%)	27 (2.63%)	6 (1.21%)	0
*Vanity Fair*	13 (12.75%)	0 (0%)	0 (0%)	0 (0%)	0
Total	102 (100%)	74 (100%)	1026 (100%)	493 (100%)	0

**Table 3 ijerph-20-04595-t003:** Number of recipes by the type of salt included in the ingredient list in eight magazines published between 2019 and 2021.

Type of Salt	Number (%) of Recipes
Canning Salt	1 (0.2%)
Celery Salt	1 * (0.2%)
Garlic Salt	3 * (0.61%)
Coarse Kosher Salt	1 (0.2%)
Coarse Sea Salt	1 (0.2%)
Fine Sea Salt	3 (0.61%)
Flakey/Flaked Sea Salt	6 (1.21%)
Fleur de Sel	1 (0.2%)
Himalayan Sea Salt	2 (0.4%)
Kosher Salt	203 (41.09%)
Onion Salt	1 (0.2%)
Salt (Unspecified)	256 (51.82%)
Sea Salt	12 (2.43%)
Season Salt	3 (0.61%)
Total	494 (100%)

* One recipe reviewed stated celery salt or garlic salt.

**Table 4 ijerph-20-04595-t004:** Number of salt types listed in the ingredient list reviewed in recipes that listed salt by magazine published between 2019 and 2021.

Magazine	Number (%) of Salt Types
*AARP the Magazine*	2 (6.66%)
*Better Homes and Gardens*	7 (23.33%)
*Cosmopolitan*	1 (3.33%)
*Good Housekeeping*	1 (3.33%)
*People*	5 (16.66%)
*Taste of Home*	10 (33.33%)
*US Weekly*	4 (13.33%)
*Vanity Fair*	0 (0%)
Total	30 (100%)

## Data Availability

Data can be found in Appendix A.

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
