# Peer review of "Salt-Containing Recipes in Popular Magazines with the Highest Circulation in the United States Do Not Specify Iodized Salt in the Ingredient List"

_ijerph, 2023, doi:10.3390/ijerph20054595_

Round 1
Reviewer 1 Report
The manuscript presents a brief report about recipes from top magazines in the US, and aims to examine the number of recipes that list salt as an ingredient and, among those, the number that specify iodized salt. A lack of emphasis on iodized salt in the magazine recipes is pointed by the authors as a fact that could be changed to help bring awareness about the importance of iodine supplementation, specially for women of reproductive age. In my opinion, the manuscript is well-written, organized, brings a relevant topic, and provides sufficient detail for the readers to understand how the study was performed. Therefore, I recommend the publication of this report. As minor considerations, I would suggest the authors to include a few words about the following topics:
1) Mention if it would be advisable or not for the US to establish mandatory salt iodization, as already performed in other countries.
2) Comment about a common recommendation for the population (including pregnant women) to reduce their salt intake, in order to prevent/control hypertension and/or edema, and how it could impact iodine nutrition even when iodized salt is preferred instead of non-iodized salt.
Reviewer 2 Report
In the recipes of journals, it is not appropriate to require using terms of "iodized salt" in stead of term "salt", although this measure might be useful for IDD prevention and control, however, it might not be their responsiblity to carry out the education of IDD prevention and control.
Reviewer 3 Report
Comments for the Author:
1. What is the percentage of iodized salt sold in the US retail market compared to non-iodized salt?
2. What is the general knowledge about iodine in health in the US public? If the answer to the question is more than average, why then do no recipes mention using iodized salt? Or the general public assumes salt that comes from the ocean would naturally have sufficient iodine?
3. Are people aware of iodine deficiency and the effect of brain damage in early childhood development if iodine is lacking in their diet?
4. Line 119 – Kosher salt is the most popular salt used in the recipes do we know why – by price or the product itself is already an iodized table salt and the authors of the recipes do not mention it as it is a piece of common knowledge locally?
5. Line 136 – iodized salt is one the major sources of iodine for the US population. It is believed that dairy products such as milk, and cheese are the major sources of iodine consumed by US consumers. There is a global trend of people consuming more plant-based milk products that contain no iodine. This is the current threat to the thyroid physiology and well beings of the vulnerable groups. That may contribute to the fall in median UIC in recent or future surveys.
6. Line 202 – ‘A greater emphasis by the media on the use of iodized salt ……’ It certainly is a way to go, regularly promoting the use of iodized salt for cooking is a novel practice. We are not here to promote the use of more salt in cooking as salt has an undeniable effect on cardiac functions if the consumption of salt is high.
7. There are unpublished reports that the sale of iodized salt has been positively linked to media emphasis on iodine nutrition. However, this effect is transient.
Round 2
Reviewer 2 Report
Although the manuscript might be helpful for prevention and control of Iodine deficiency disorders, I personnally thought there might be some bias for journals selection, and it will be very difficult to require journals using the word "iodizsed salt" instead of "salt", as a result, the significance of the manuscript will be discounted.
Author Response
Although the manuscript might be helpful for prevention and control of Iodine deficiency disorders, I personnally thought there might be some bias for journals selection, and it will be very difficult to require journals using the word "iodizsed salt" instead of "salt", as a result, the significance of the manuscript will be discounted.
Authors’ response:
Thank you for your feedback.
We completed searches of popular magazines, not scientific journals. Specifically, we aimed to search the 10 most popular magazines in the US based on circulation figures. However, two of the magazines were not available in the libraries that we searched. We do not believe the exclusion of these two magazines systematically biased the research.